# Feeding Ecology of Reintroduced Golden Parakeets (*Guaruba guarouba*, Psittacidae) in a Protected Area in the Amazon Forest

**Marcelo Rodrigues Vilarta** [1,2,3,*], **Thaís Tamamoto De Moraes** [3], **Maria Fernanda Naegeli Gondim** [3,4,5], **Crisomar Lobato** [6], **Mônica Nazaré Rodrigues Furtado Da Costa** [6], **Rubens de Aquino Oliveira** [6] and **Luís Fábio Silveira** [1,3]

1. Seção de Aves, Museu de Zoologia da Universidade de São Paulo, Ipiranga 04263-000, Brazil; lfs@usp.br
2. Programa de Pós-Graduação em Zoologia, Departamento de Zoologia, Universidade de São Paulo, Ipiranga 04263-000, Brazil
3. Lymington Foundation, Estrada Dias Gomes 2.200, Juquitiba 06950-970, Brazil; thaisfundacaolymington@gmail.com (T.T.D.M.); veterinarialymington@gmail.com (M.F.N.G.)
4. Zooparque de Itatiba, Rod. Dom Pedro I km 95, Itatiba 13254-762, Brazil
5. Departamento de Medicina Veterinária, Universidade São Francisco, Avenida Senador Lacerda Franco, 360, Itatiba 13250-400, Brazil
6. Instituto de Desenvolvimento Florestal e da Biodiversidade do Estado do Pará, IDEFLOR-Bio, Avenida João Paulo II, Belém 66610-770, Brazil; crisomarlobato@yahoo.com.br (C.L.); mfurtado.bio@gmail.com (M.N.R.F.D.C.); rubens.aquino25@gmail.com (R.d.A.O.)
* Correspondence: marcelovilarta@hotmail.com

**Abstract:** The Golden Parakeet is an endemic Brazilian flagship species that has suffered from poaching and habitat loss, leading to local extinctions in the urbanized parts of the Amazon. We reintroduced six groups of mostly captive-bred parakeets in a protected area. The birds were acclimatized for at least five months at the release site, where they were trained to recognize native foods and develop foraging skills. Subsequently, we conducted a soft release, followed by daily supplementation and monitoring. For three years following the release we recorded their diet, feeding behavior, and how they adapted to wild foraging. The reintroduced birds fed on 23 plant species, with 13 not being previously listed in past studies. The three most consumed species corresponded to 77% of their feeding records. Parakeets spent more time feeding in altered landscapes and secondary vegetation than in the preserved forest. Most of the feeding happened during the rainy season when most of their favorite plants are fruiting. The parakeets' incorporation of new species in their diet and their transition from supplemental to natural feeding happened gradually, as we did not reduce food offerings. Parakeets that showed site fidelity were able to find native food rapidly following release, but individuals that dispersed immediately had more difficulty finding food. This study showed that captive-bred Golden Parakeets can transition to a wild diet following a gradual reintroduction process.

**Keywords:** captive-bred; diet; foraging; reintroduction; site-fidelity; monitoring; supplementation



## 1. Introduction

Information on a species' diet and feeding ecology is important for predicting how landscape changes may affect species and for designing effective conservation programs and management actions aimed at population recovery [1,2]. Reintroductions have been implemented to restore populations to areas in which they had been extirpated, and these initiatives have gained popularity as a result of their media appeal and symbolic meaning as a correction to human-driven defaunation [3]. However, many endeavors end up being unsuccessful due to the plethora of biological and methodological obstacles involved [4]. The challenge is considerably higher when using captive-bred animals instead of captured wild individuals since they are mostly unfamiliar with the native environment and lack essential learned behaviors, such as foraging skills [5,6].

Due to their vibrant appearance and conspicuous nature, parrots are considered highly charismatic fauna. This trait also leads to their endangerment by making them valuable targets for poachers and traffickers [7]. About one third of all parrot species are threatened with extinction worldwide, making them one of the most threatened groups of birds [8,9]. Brazil has a leading part in this problem as it has both the largest number of psittacine taxa and endangered species [9,10]. Since these birds are highly forest-dependent (70%) and the country still faces alarming rates of deforestation, parrots are likely to face even more severe declines shortly [9,11,12]. Despite this dire outlook, the group is highly represented in conservation actions that aim to reverse or compensate for the local extinctions that are already in progress [13].

Reintroductions are still incipient in Brazil, but recent years have seen an increasing number of projects restoring extirpated populations of parrots, ranging from non-threatened species (*Amazona aestiva*) [14] to endangered ones such as *Amazona vinacea* and *Anodorhynchus leari* [15,16], with even a species that was extinct in the wild (*Cyanopsitta spixii*) being brought back [17]. Despite being a favorite flagship species for conservation efforts, some parrots have been insufficiently studied, leaving substantial gaps in information on their habitat and foraging requirements, which are necessary for supporting reintroduction efforts and increasing their chances of success.

The Golden Parakeet (*Guaruba guarouba*) is an exuberant parrot endemic to the Brazilian Amazon Rainforest. Its lush plumage has the green and yellow colors of the national flag, making it a prime candidate for the Brazilian national bird [18]. The species has suffered dramatically from the illegal pet trade, and with most of its wild population distributed across the most threatened areas in the Amazon, it is listed as vulnerable to extinction [8,19]. Few authors have thoroughly studied the species despite its symbolic importance and concerning status. Oren and Novaes [20] described the parakeets' biology, while Silveira and Belmonte [21] provided information on their reproductive behavior. Laranjeiras [22] and Laranjeiras and Cohn-Haft [19] described the known geographic distribution of the species and estimated its population size. However, no study has focused on its diet and feeding ecology, which are mostly known from opportunistic observations.

In 2017, the first initiative to reintroduce Golden Parakeets from captivity was implemented in a protected area located in the capital of Pará, Belém, where the species was deemed extinct for over a century [23,24]. The reintroduction process is ongoing and has already established a sizeable population in the region, which is currently being monitored. Identifying and conserving key food resources and habitats is vital to the long-term conservation of parrot communities [1]. In this study, we aim to describe the diet of reintroduced Golden Parakeets, their foraging habits and habitat, and the process of their post-release transition from supplemental to native feeding.

## 2. Materials and Methods

The reintroduction and most of the monitoring was conducted inside the Utinga State Park, an Integral Protection Unit located in the metropolitan region of Belém, capital of the state of Pará. The protected area is 13.93 km$^2$ and connected to another protected area to the east and south but also bordered by urban areas to the north and west.

The park landscape is mainly composed of primary rainforest (60%), with both terra firma and Igapó (flooded) physiognomies, as well as altered landscapes and secondary vegetation (15%) resulting from previous illegal occupation and management interventions. The remaining 25% of the area is covered by two large bodies of water and aquatic vegetation [25,26]. As in most of the Amazon Rainforest, the seasonality is described as having two periods: the rainy season (December to May) and the dry season (June to November). Its average monthly precipitation is 274 mm, but significant variations range from 44 mm to 743 mm [26].

The birds selected for reintroduction were born in captivity at the Lymington Foundation (LF), a conservationist organization that is dedicated to preserving Brazil's native fauna, and which operates a captive breeding facility for endangered parrots. At the LF,

groups of 10 individuals were prepared each year by selecting the parakeets that exhibited a good capacity for flight and socialization. They were screened for common psittacine diseases (herpesvirus, bornavirus, circovirus, mycoplasma, chlamydia) according to the local regulations governing animal translocations.

Five groups were sent to Belém via plane (November 2018; June 2021; January 2022; November 2022; June 2023) and placed in an acclimatization aviary built at the release site. The aviary was built on a previously cleared site of secondary vegetation, surrounded by tall trees. The aviary structure is composed of two modules: a maintenance enclosure of 6 m × 2 m × 2 m connected to a larger one of 5 m × 5 m × 5 m. The parakeets were kept in the aviaries for at least 5 months before release. During the acclimatization period, they were presented to potential predators both directly, by bringing live boa snakes inside the enclosure to startle them, and indirectly, as birds of prey that were naturally present in the area and could occasionally be seen diving for small animals in their proximity. They also received daily training to improve their flight strength, by stimulating their movement between enclosures, and were presented with native food.

Their diet was gradually shifted from the standard commercial food that the birds were already familiar with in captivity (papaya, apple, banana, mango, extruded ration) to the native fruits found in the park that were previously known to be included in their natural diet (*Byrsonima crassifolia*, *Euterpe oleracea*, *Chrisobalanus icaco*, *Anacardium occidentale*, *Psidium guajava*, *Inga edulis*). Whenever possible, the native food items were served attached to whole parts of their respective plants in various spots inside the enclosures, training the parakeets to develop naturalistic foraging behavior. Birds would only be considered ready for release once they could be observed correctly processing and feeding on these fruits.

In January 2021, after the acclimatization period, we started reintroducing the current group of Golden Parakeets through a soft release. In our approach, we released the birds individually or in pairs over the course of one to three months, prompting them to return to the aviary daily to avoid dispersal. For that, we offered constant food supplementation in three suspended feeders near the aviary without reducing their availability over time. We favored using native food in the feeders, but we also had to complement it with commercially available fruits and extruded ration for the parrots in order to provide the released birds with a continuous supply of food. We also placed nest boxes in the area to allow a safe option for roosting. Once the release process of a group of birds was finished, the following group was brought to the aviary as soon as possible, serving as social attractants. Four additional groups were released in January 2022, September 2022, May 2023, and January 2024, respectively.

Foraging events were recorded during weekly monitoring campaigns and opportunistic observations from April 2021 to February 2024. During the campaigns, the groups of released parakeets were searched for and followed throughout the day, from when they left their nest boxes, which they used as dormitories, in the early morning until they returned to them at the end of the day. Whenever the birds could not be found by hearing their vocalizations or visualizing them, we used telemetric tracking and playback calls to locate them. If they could not be located for a long period the campaign was stopped. Once the birds were located, we maintained constant observation using 8 × 42 binoculars and photographic cameras with 70–300 m lens zoom to record any feeding events. For each feeding observation, we noted the plant species and the parts that were being consumed on-site. For unknown species, we collected branches and parts discarded by the parakeets for later identification. For each feeding bout, the location's habitat type was classified according to its habitat characteristics or location on the park map [26]. The duration of the feeding event was timed from the moment at least one group member began eating and ended when none of them were eating or when the group suddenly flew away. We only considered feeding events valid when the birds could be observed actively eating and not just biting and immediately discarding plant parts after tasting them.

## 3. Results

We were able to collect foraging data for 86 of the 164 observation campaigns that we conducted; in the remaining cases we lost track of the birds or could not reliably observe their activity to make feeding records. In the successful campaigns we accumulated 1032 h of search efforts in which we recorded 134 h of natural feeding observations and a total of 429 feeding events. Feeding sessions took 19.7 min on average, with a standard deviation of 18.4 min.

We recorded the released Golden Parakeets foraging on 23 plant species from 14 families in the wild (Table 1). Twenty-one were native species, and two were exotic. Some species were consumed much more frequently than others. The three most eaten species (*Byrsonima crassifolia*, *Euterpe oleracea*, *Tapiria guianensis*) represented more than 77% of the feeding time and 65% of feeding bouts. While most plant families were represented by one species, Arecaceae had the most representatives, with six species, and then Anacardiaceae with three species.

**Table 1.** Plant species eaten by the released Golden Parakeets at the Utinga State Park and in its vicinity. Asterisks (*) refer to species that were not previously reported in the literature. Part: plant parts that were consumed. Bouts: number of feeding sessions. Time: total minutes the parakeets spent feeding on the plant. %: proportion of the total time spent feeding on the given species. J–D: months in which the plant was consumed. Fr: fruit; fw: flower; sd: seed. X: indication of the plant being consumed.

| Family | Species | Part | Bouts | Time (min) | % | J | F | M | A | M | J | J | A | S | O | N | D |
|---|---|---|---|---|---|---|---|---|---|---|---|---|---|---|---|---|---|
| Anacardiaceae | *Tapirira guianensis* | Fr, sd | 57 | 1510 | 18.8 | X | X | X | | | | | | X | X | X | X |
| | *Mangifera indica* | Fr | 2 | 28 | 0.3 | X | | | | | | | | | | X | |
| | *Anacardium occidentale* | Fw, fr | 2 | 15 | 0.2 | | | | | | | | X | | | X | |
| Araliaceae | * *Scheflera morototoni* | Fr, sd | 3 | 40 | 0.5 | | | | X | X | | | | | | | |
| Arecaceae | *Euterpe oleracea* | Fr, sd | 79 | 1590 | 19.8 | X | X | X | X | X | X | X | X | X | X | X | X |
| | * *Elaeis guineensis* | Fr | 16 | 320 | 4.0 | X | X | | | | | | | | X | X | X |
| | * *Mauritiella armata* | Fr, sd | 8 | 145 | 1.8 | | | | X | X | | X | X | | | | |
| | * *Socratea exorrhiza* | Fr, sd | 4 | 80 | 1.0 | | X | X | | | | | | | | | |
| | *Oenocarpus bacaba* | Fw, fr | 5 | 60 | 0.7 | | | | | | | | X | | | | |
| | * *Mauritia flexuosa* | Fw, fr, sd | 7 | 150 | 1.9 | | | | | | | | | X | X | | |
| Burseraceae | * *Trattnnickia* sp. | Fr, sd | 3 | 25 | 0.3 | | | | | | X | X | | | | | |
| Chrysobalanaceae | * *Chrisobalanus icaco* | Fw, fr | 32 | 240 | 3.0 | X | | | | X | X | | | | X | X | X |
| Clusiaceae | *Symphonia globulifera* | Fw, fr, sd | 8 | 124 | 1.5 | | | | X | X | | | | | | | |
| Cyperaceae | *Rhynchospora cephalotes* | Fr, sd | 9 | 40 | 0.5 | | | | | | | | X | X | X | X | |
| Dilleniaceae | * *Tetracera* sp. | Fr, sd | 2 | 14 | 0.2 | | | | | | | | X | | | | |
| Erythroxilaceae | * *Erythroxilum* sp. | Fr, sd | 2 | 25 | 0.3 | X | X | | | | | | | | | | |
| Fabaceae | *Ingá edulis* | Fr, sd | 2 | 24 | 0.3 | | | | X | | | | | | | | |
| Hypericaceae | * *Vismia guianensis* | Fw, fr, sd | 5 | 31 | 0.4 | | X | X | X | | | X | | | | | |
| Malpighiaceae | *Byrsonima crassifolia* | Fw, fr, sd | 140 | 3115 | 38.7 | X | X | X | X | | | | | X | X | X | X |
| Melastomataceae | *Miconia cuspidata* | Fr, sd | 14 | 190 | 2.4 | | | | X | X | X | X | X | | | | |
| | *Miconia prasina* | Fr, sd | 13 | 60 | 0.7 | | X | | | | | | X | X | | X | X |
| Myristicaceae | * *Virola surinamensis* | Sd | 8 | 175 | 2.2 | | | | X | X | X | | | | | | |
| Onagraceae | * *Ludwigia decurrens* | Fw | 8 | 42 | 0.5 | | | | | | | | | X | X | X | |

Golden Parakeets mainly fed on seeds and fruits but were occasionally seen eating buds and flower parts. They typically moved toward the terminal end of twigs and branches to pluck fruit, and then processed the fruit while holding it in one foot and hanging from

the perch with the other. Often, they could be seen in varied and awkward positions, such as upside down, while still being able to hold their food efficiently (Figure 1a). This behavior could be observed even before their release; when the birds processed the native food that was offered they would rapidly exhibit this ability.

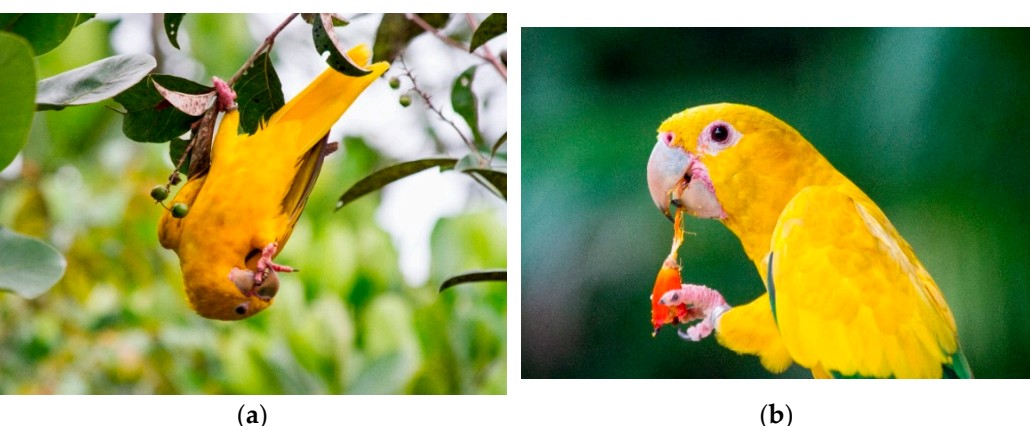

(**a**)                                                                                 (**b**)

**Figure 1.** (**a**) Released Golden Parakeet feeding on unripe *Byrsonima crassifolia* by perching with one foot while processing the fruit and seed with the other. (**b**) Released Golden Parakeet feeding on *Elaeis guineensis* by peeling the oily fruit whilst holding it with its foot.

While the parakeets were sometimes observed eating the pulp of fruits, their most common feeding method was to peel most of the pericarp until only the seed was left and then break it and macerate the endosperm into smaller pieces to ingest and lick them. In the process, the parakeets fed only on a small portion of the food items, discarding the rest. Overall, the parakeets acted as seed predators, using these resources regardless of the fruit's ripeness stage. For the seeds that were too hard to access with their beak (*Elaeis guineensis*, *Mauritia flexuosa*), the parakeets would only peel the oily pericarp and lick its contents (Figure 1b).

While foraging in sections of primary forest with tall trees, the parakeets would usually land on top of the canopy, searching for edible parts, rarely descending into the lower parts of the dense vegetation. This would only be observed when they had to reach the palm fruits that grow under the canopy. Conversely, when feeding on the edges of the forest and in secondary vegetation patches, they would often maneuver to the relatively lower parts of plants until they found the fruits, leaving them more exposed. Whenever a group would gather to feed, one or more individuals would act as sentinels and remain positioned in a higher part of the canopy or on light poles and power lines along the roads. The sentinels would stay vigilant and not feed until the group moved to the next spot, where other individuals would take their place (Figure 2).

Although we have never observed the Golden Parakeets feeding on the ground outside the release site, this behavior was frequently exhibited in the vicinity of the enclosure and the suspended feeders, where their favorite food items would fall to the ground. This was also the only area where they were seen feeding on herbaceous species (*Ludwigia decurrens*, *Rhynchospora cephalotes*). This same behavior was observed during their acclimatization period, as the birds would forage in the lower parts of the aviary after the feeders were depleted.

During the observation period, the Golden Parakeets were seen eating in the wild mainly during the morning, from 7:00–11:00, and then from 16:00–17:00. In the period of 12:00–15:00 the parakeets would mostly perch in taller trees or in the vicinity of the aviary and rest until resuming their search for food. The group would usually cease their activities and return to the nest boxes around 18:00, though on days with late sunsets, they could be seen feeding past 18:00 (Figure 3).

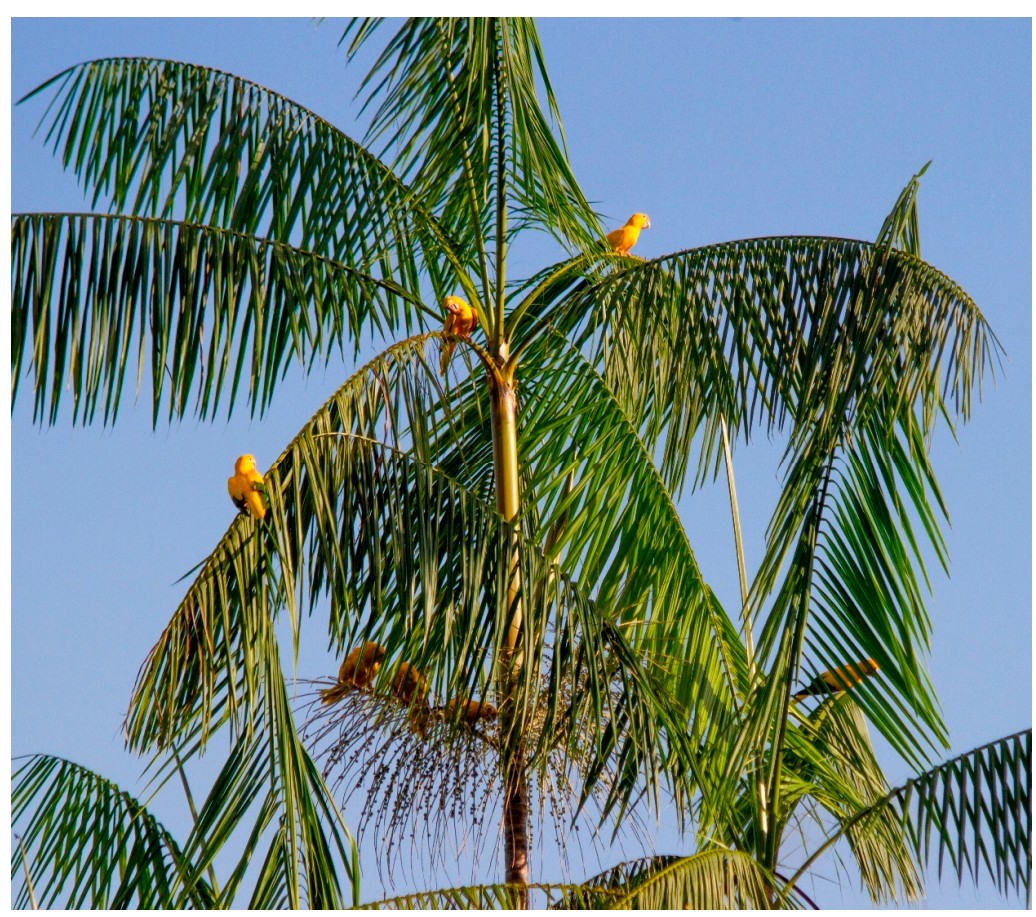

**Figure 2.** Group of released Golden Parakeets foraging on a *Euterpe oleracea* palm. While three individuals feed on the fruits, four others remain vigilant at the top.

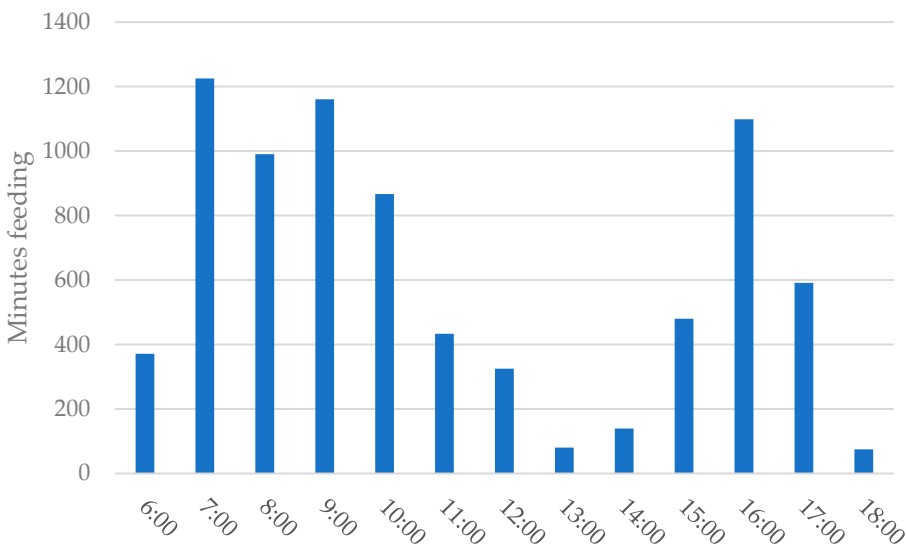

**Figure 3.** Hourly distribution of feeding times of the released Golden Parakeets.

The increase in the species diversity in the diet of the reintroduced parakeets was relatively gradual (Figure 4). The four most consumed species were also the first to be recorded following the birds' release. The top two (*Byrsonima crassifolia*, *Euterpe oleracea*) were the species that we most often provided during the food training due to their availability. However, the third and fourth (*Tapirira guianensis* and *Elaeis guinensis*) were not initially

presented during the acclimatization period, but were rapidly found by the parakeets and incorporated into their diet once in the wild. Four species frequently offered during the food training (*Psidium guajava*, *Anacardium occidentale*, *Mangifera indica*, *Syzygium malaccense*) were seldom or never explored in the wild, despite their availability in the study area.

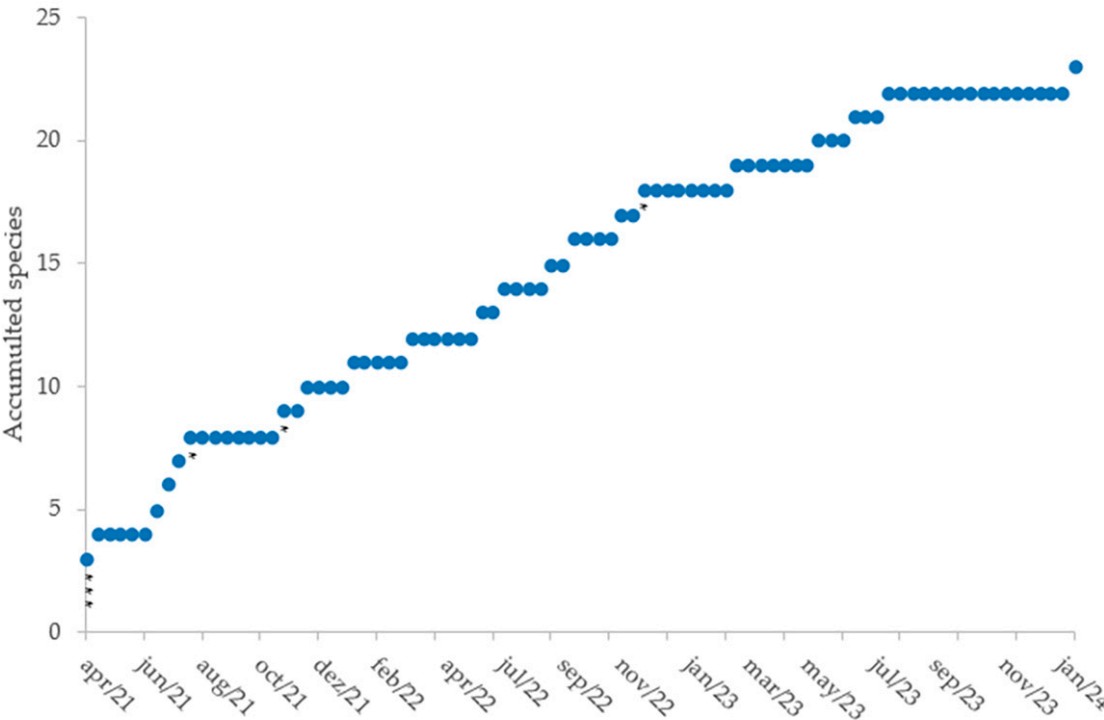

**Figure 4.** Plant species accumulation curve in the diet of reintroduced Golden Parakeets, as documented during post-release monitoring campaigns. Each asterisk (*) refer to the first time a plant species provided in food training was used in the wild.

The group of released parakeets gradually increased its natural feeding over three years. During the first months after their release, almost all their feeding time was spent at the suspended feeders, and during the first year parakeets typically spent over 70% of their foraging time feeding on supplemental food. In the second year, we observed the parakeets spending around half of their time at both natural and supplemental sources, and by the third year most of their feeding was carried out in the wild (Figure 5). Four additional releases were made from January 2022 to December 2023. We observed that once the newly released individuals started following the established group, their feeding behavior quickly converged to what the experienced birds were already doing. However, it was still possible to observe new individuals separating from the group in order to return and feed on supplemental food in their first weeks in the wild.

One individual that dispersed immediately after release was observed to return alone to the release site after 15 days, demonstrating that some Parakeets may survive without supplemental feeding, even after immediately dispersing post-release. However, four individuals who dispersed immediately after release were followed for two days each, and during this period we did not observe any acts of natural feeding. We recaptured three individuals to prevent their starvation, attracting them with feeders and sunflower seeds. The fourth individual entered a house and was searching for human food, allowing us to recapture it. Another individual dispersed after spending one year with the group, and was resighted nine months later with a group of *Psittacara leucophtalmus*.

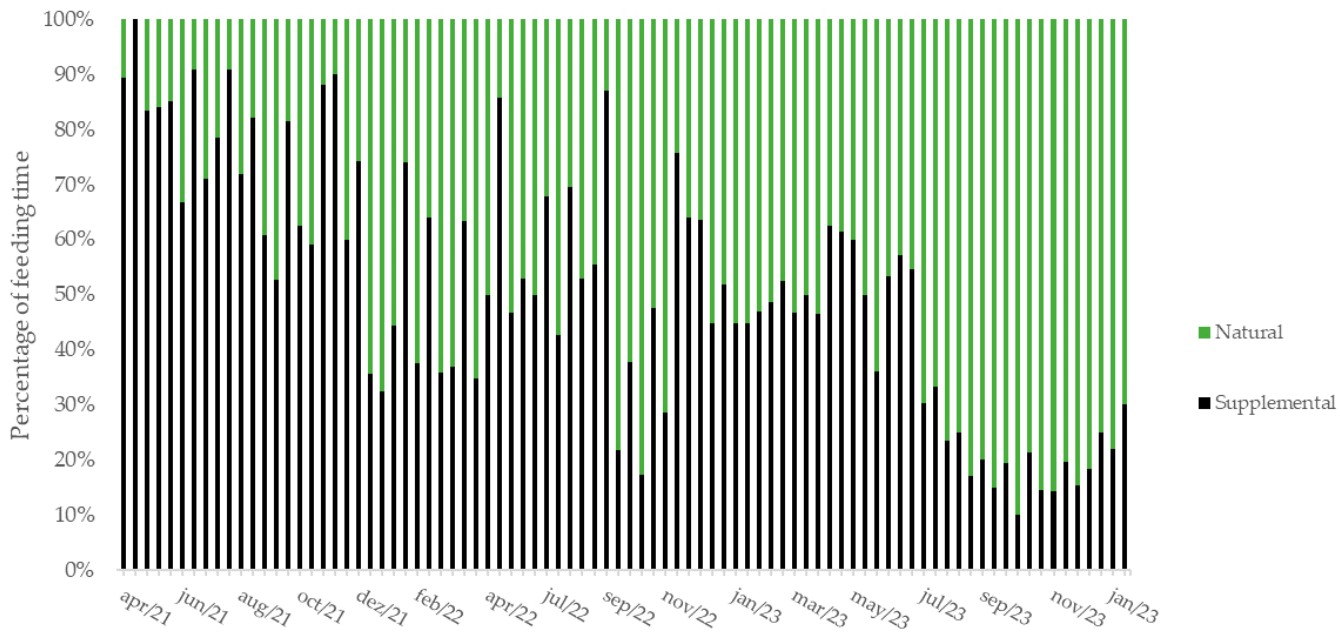

**Figure 5.** Percentage of time that the group of released parakeets spent feeding in the wild (Natural) or at suspended feeders (Supplemental) throughout the monitoring period.

Golden parakeets fed mostly on vegetation located in altered landscapes and secondary forests (Table 2), amounting to almost 80% of their observed feeding time, while, in the primary forest, the amount of feeding time was only 20% (Figure 6a). Most plant species were consumed in terra firma environments (Table 2). Their observed feeding time was greater during the rainy season (63%) than during the dry season (37%) (Figure 6b). *Euterpe oleracea* was the only species consumed throughout the year. Although *E.oleracea*'s fruit production is greater during the wet season, parakeets were able to find and consume unripe fruits during the dry season. The parakeets consistently spent much less time feeding in primary vegetation than in secondary vegetation, but during the dry season the percentage of their time feeding in primary vegetation nearly doubled (Figure 7a,b).

**Table 2.** Further details on the plant species eaten by the Golden Parakeets and the respective environments where they were found. Asterisks (*) refer to the plant species that were initially provided during food training. Vegetation type: predominant physiognomy where the plant was eaten. Forest Formation: terrain type where the species was predominantly used. Class: morphological type of the species. Common name: regional names that are used to refer to the species. Further information on the park flora may be found in Ferreira et al. [25].

| Family | Species | Vegetation Type | Forest Formation | Class | Common Name |
|---|---|---|---|---|---|
| Anacardiaceae | *Tapirira guianensis* | Secondary | Terra firma | Arboreal | Pau pombo |
| | * *Mangifera indica* | Secondary | Terra firma | Arboreal | Manga |
| | * *Anacardium occidentale* | Secondary | Terra firma | Arboreal | Cajú |
| Araliaceae | *Scheflera morototoni* | Primary | Terra firma | Arboreal | Morototó/mandiocão |
| Arecaceae | * *Euterpe oleracea* | Secondary | Terra firma and flooded | Palm | Açaí |
| | *Elaeis guineensis* | Primary | Terra firma | Palm | Dênde |
| | *Mauritiella armata* | Primary | Terra firma and flooded | Palm | Miriti/Buritirana |
| | *Socratea exorrhiza* | Primary | Terra firma | Palm | Paxiubinha/Palmeira andante |
| | *Oenocarpus bacaba* | Primary | Terra firma | Palm | Bacaba |
| | *Mauritia flexuosa* | Primary | Terra firma | Palm | Buriti |

**Table 2.** *Cont.*

| Family | Species | Vegetation Type | Forest Formation | Class | Common Name |
|---|---|---|---|---|---|
| Burseraceae | *Trattinichia* sp. | Secondary | Terra firma | Arboreal | - |
| Chrysobalanaceae | * *Chrisobalanus icaco* | Secondary | Terra firma and campinarana | Shrub | Ajirú |
| Clusiaceae | *Symphonia globulifera* | Primary | Terra firma | Arboreal | Anani |
| Cyperaceae | *Rhynchospora cephalotes* | Secondary | Terra firma | Herb | - |
| Dilleniaceae | *Tetracera* sp. | Secondary | Terra firma | Shrub | - |
| Erythroxilaceae | *Erythroxilum* sp. | Secondary | Terra firma | Shrub | - |
| Fabaceae | * *Inga edulis* | Secondary | Terra firma | Arboreal | Ingá |
| Hypericacea | *Vismia guianensis* | Secondary | Terra firma and capinarana | Arboreal | Lacre |
| Malpighiaceae | * *Byrsonima crassifolia* | Secondary | Terra firma and campinarana | Arboreal | Muruci |
| Melastomataceae | *Miconia cuspidate* *Miconia prasine* | Secondary Secondary | Terra firma Terra firma | Arboreal Shrub | Pixiricão Pixirico |
| Myristicaceae | *Virola surinamensis* | Primary | Terra firma | Arboreal | Ucuuba |
| Onagraceae | *Ludwigia decurrens* | Secondary | Terra firma | Herb | - |

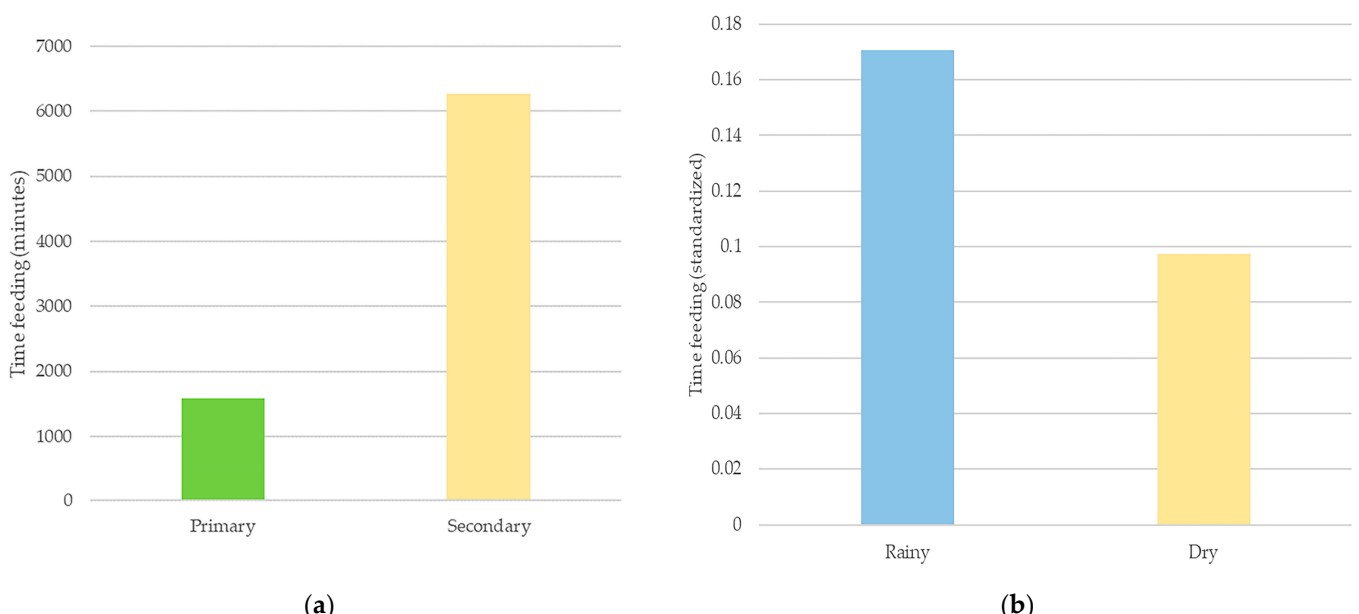

(**a**)　　　　　　　　　　　　　　(**b**)

**Figure 6.** (**a**) Time the released parakeets spent feeding on primary or secondary vegetation types. The park is mostly covered by primary vegetation, 60%, while secondary only covers 15% of the area. (**b**) Feeding time (observed feeding/total effort) of the released parakeets during the rainy and dry seasons.

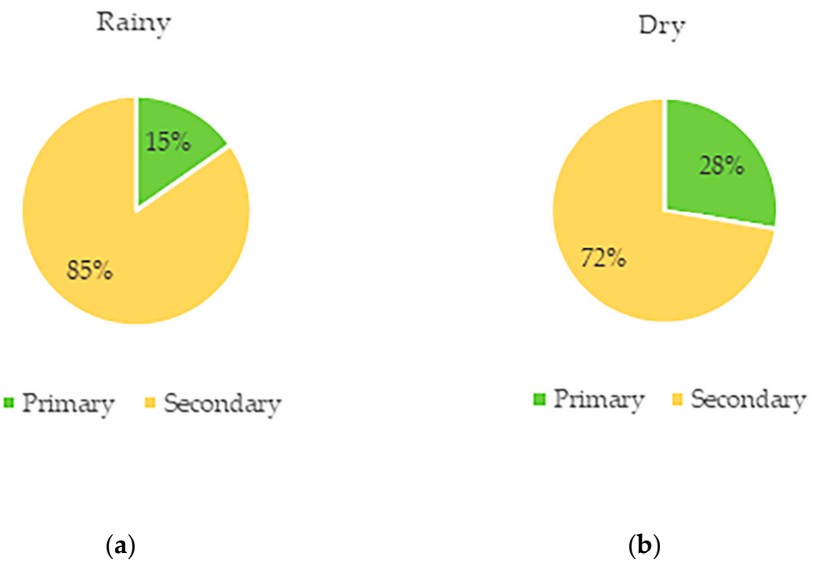

(**a**)           (**b**)

**Figure 7.** (**a**) Relative distribution of parakeets' feeding time during the rainy season between primary and secondary vegetation. (**b**) Relative distribution of their feeding time during the dry season between primary and secondary vegetation physiognomies.

## 4. Discussion

The number of species consumed by reintroduced Golden Parakeets was within the range reported by Renton et al. [27], who reviewed dietary studies of parrots, showing they would feed on 7 to 26 plant species on average. Ten species that compose the reintroduced Parakeets' diet have also been reported in the feeding observations of wild Golden Parakeets [18,20,22], whereas thirteen species had not been previously listed. The 13 new species account for only 16.6% of the feeding records, while the previously reported species accounted for a much larger share of those records (83.4%).

*Byrsonima crassifolia*, which was the most frequently consumed food item in our study, was also reported by Laranjeiras [22], who made most of the previous observations of Golden Parakeets' foraging. The fruiting period of *B. crassifolia* overlaps with the parakeets' breeding season (December to February); even so, we recorded the parakeets feeding on it when the first unripe fruits appeared in September. These observations indicate that this fruit is likely important for the Golden Parakeets, and its availability should be considered when reintroducing a population in a given area. *B. crassifolia* is the most typical representative of the "campinaranas" phytophisiognomy in the study area [25], showing that, despite not being suitable for roosting and nesting, these opened areas are favorable for supplying food for Golden Parakeets.

The second most consumed species on our list, *Euterpe oleraceae*, which was consistently consumed throughout the year in our study, was previously reported by Sick [18] but in no other published reports. The açaí palm, despite being found in all of the Amazon, is densely present in urban environments and neighboring forests due to its commercial value. Additionally, the açaí fruit was one of the food items that was most often provided during food training, and this might explain why it was so frequently consumed post-reintroduction.

The Arecaceae family was the most represented, with six species consumed by the parakeets. Palms, which produce fat-rich fruits that are available year-round, are thought to be essential food sources for parrots, many of whom consume fruits from multiple palm species [27,28]. Despite being mainly seed predators, psittacines are known to form mutualist dispersal networks with palms particularly [29,30]. Our observations were consistent with this, as we saw multiple instances of stomatochory dispersal with palm fruits, mainly *Elaeis guineensis* and *Euterpe oleracea*, but only rarely with other species. These dispersal events mainly occurred when the group was flushed during feeding events after spotting a predator. Some parakeets would fly off still holding the fruits and seeds in

their beak, as also reported by Carrete et al. [29], Sazima [31], and Silva et al. [32] on other psittacines. The latter were also seen carrying the fruits with their feet during flight, which we did not observe with these parakeets.

As reported for other studies of parrots, we observed that Golden Parakeets wasted a great deal of food during processing and feeding. While this could be explained by a high degree of selectivity for particular plant parts, it could also result from inefficient food processing and the frequent dropping of food while feeding [33,34]. The latter explanation seems more probable, considering this was also observed during captivity, where the ever-constant food supply would not pressure the birds to learn caution with resource handling.

Sentinel behavior, on the other hand, is rarely documented for parrots. However, it has already been reported in *Psittacus erithacus* [35] and *Anodorhychus leari* [36], so the rapid display of this behavior in reintroduced parakeets, without previous teaching by wild individuals, was an excellent indication of their capacity for developing wild behaviors and avoiding predation. On the other hand, the retention of inappropriate behaviors such as ground foraging, which has never been observed in wild individuals, also reflects that some captivity-related problems may persist after the birds' reintroduction.

The order in which food plant species were incorporated into their diet post-reintroduction was apparently influenced by pre-release training; three species offered in the aviary were among the first to be consumed post-reintroduction (Figure 4). However, they still quickly explored unknown species that would later be considered important resources. This showed that food recognition training was important and influential in the first few weeks after release, and the Golden Parakeets' innate foraging abilities also played an essential role in shaping their food repertoire.

Although the parakeets in our study were observed feeding on almost all species reported in previous studies, *Croton matourensis*, which has been listed as an important resource for wild parakeets [37], has not been used by the reintroduced birds to date, even though it is present at the study site. We should note that we were unable to collect *Croton matourensis* for food training and supplementation. *Psidium guajava*, which is also known to be a favorite psittacine resource [38,39], was also never consumed in the wild but was offered daily in the supplemental feeders.

The fact that some plants have not yet been explored by the released birds and that the accumulation curve was not completely flattened suggests that the diet of golden parakeets may be even richer, and that new species are yet to be discovered and incorporated. This is an indication that the direct observation method may not be enough to provide a complete list of the species' diet, given the limitations imposed by the forest terrain that prevented us from visualizing the birds most of the time. Our observations contrasted with Volpe et al. [39], who reported a flattening of the curve in less than one year of monitoring released *Ara chloropterus*. The slower discovery rate of new items by the Golden Parakeets and the lengthy adaptation process from supplemental to native food may result from the soft release approach we chose, providing constant supplementation as an incentive to keep the reintroduced birds closer to the release site. While our approach was expected to promote their slower adaptation to the wild, it was also a solution to reduce the elevated rate of early dispersal that hampered the monitoring of the parakeets at the beginning of the project [24].

The observation of released individuals who could not forage in the wild in their earliest days post-release raised concerns about the dispersed parakeets that could not be monitored. Even with intensive feeding training, we observed that the release process can be difficult for some individuals who are unable to forage successfully, especially when they become separated from the group. Although we do not know for certain whether these individuals would ultimately starve if not recaptured, we highlight the importance of promoting site fidelity in conjunction with supplemental feeding for Golden Parakeets to allow for a safer and more successful transition from the aviary to the wild. The first year post-reintroduction showed that food supplementation provided a safe and reliable foraging option for reintroduced Golden Parakeets. The importance of food supplementation

has been highlighted by other studies, citing it as a critical factor contributing to successful psittacine reintroductions [7].

Even though the release area was mostly covered by primary forest, the parakeets clearly preferred to forage in disturbed areas of secondary vegetation (Figure 6a), presumably because their three most consumed species are found in human-altered areas. Those species fruit primarily during the rainy season, which explains their greater use of disturbed areas during the wet season. When those fruits are not available during the dry season, the parakeets then shift to feeding more in primary vegetation habitats.

Considering wild food availability, our data suggest that the ideal period for releasing Golden Parakeets would be between the final months of the dry season and the beginning of the rainy season, when the reintroduced birds have a higher chance of locating abundant food. However, the beginning of the rainy period also coincides with the breeding season of Golden Parakeets, which we previously considered an inappropriate time to release the birds, given the higher tendency for territorial disputes that may hamper the inclusion of new individuals into the group [24].

This study showed that, despite the challenges imposed by their origin, captive-bred Golden Parakeets can adjust to a natural environment given a supportive reintroduction process and can develop most of the expected foraging behaviors of a wild parrot. This is critical in establishing a self-sustaining population at the study site.

**Author Contributions:** Methodology, L.F.S. and M.R.V.; funding and project administration, L.F.S., C.L., M.N.R.F.D.C. and R.d.A.O.; fieldwork M.R.V.; exams, diet planning, and preparation of the Golden Parakeets, T.T.D.M., M.F.N.G. and M.R.V.; data processing M.R.V.; writing and reviewing, M.R.V., L.F.S., C.L., M.N.R.F.D.C., R.d.A.O., T.T.D.M. and M.F.N.G. All authors have read and agreed to the published version of the manuscript.

**Funding:** This research was funded by Instituto de Desenvolvimento Florestal e da Biodiversidade do Estado do Pará, IDEFLOR-Bio and supported by BluestOne and the Parrot Wildlife Foundation.

**Institutional Review Board Statement:** This study was approved by the Ethics Committee of the Instituto de Biociências da Universidade de São Paulo (protocol code 227/2015, approved in 30 June 2015) and has a Federal license issued by ICMBio, number 56616-11.

**Informed Consent Statement:** Not applicable.

**Data Availability Statement:** Data are contained within the article.

**Acknowledgments:** This project was enabled by many people and institutions. We thank William Wittkoff, Linda Wittkoff, Luiz Carlos Oliveira, Nancy Fragnan, and Vinicius Lopes for maintaining the Lymington Foundation's operations, ensuring the basis of this project. Rosemary Low and Roelant Jonker, thank you for reviewing and providing insights throughout the project. We thank the staff at IDEFLOR-Bio, Ana Claudia Aranha, Ana Paula Rodrigues, Neusa Renata Emin, Nívia Pinto, Luiziana Moura, Camila dos Anjos, Sindomar Cardoso, Tainá Garcia, and the interns, for their technical and operational support. Arnaldo Algaranhar, Ellen Eguchi, and Camilo Gonzalés for the veterinary support. Roberto Veloso, Juvenal Amaral, and Tarcísio Rodrigues for providing more birds. Leandro Ferreira Vale, José Leonardo Magalhães, Carlos Boelter, and José Aluísio Fernandes for the support in identifying plant species. Augusto Jarthe and João Victor for their help with herpetological management and Golden Parakeet sightings in the park. Marcelina Parreira for field support during releases. Alexandre Resende, Guilherme de Oliveira, Ananda Porto, and Cristiane Prizibisczki for further elevating the media projection of this work. Neiva Guedes and Donald Brightsmith for the thorough reviewing process of this paper.

**Conflicts of Interest:** The authors declare no conflicts of interest.

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
