# Peer review of "Feeding Ecology of Reintroduced Golden Parakeets (Guaruba guarouba, Psittacidae) in a Protected Area in the Amazon Forest"

_diversity, doi:10.3390/d16030188_

Round 1
Reviewer 1 Report
Comments and Suggestions for Authors
Parabéns aos autores, fizemos um bom trabalho de reintrodução numa área onde um foco específico já estava extinto. Fizeram uma pesquisa bem planejada e acompanharam monitorando a dispersão e alimentação dos indivíduos liberados. Boa análise e boas conclusões. Parabéns!!!
Reviewer 2 Report
Comments and Suggestions for Authors
This paper presents data on the foraging diet of captive-bred Golden Conures following their release to the wild, and a description of the process used to aid their transition from feeding in captivity to foraging independently on plant species available in the release area. I found the manuscript to be generally well-organized and well-written, and my comments and suggestions are mainly offered to improve the readability and flow of the text. Some clarification of how feeding times were quantified needs to be provided/revised (e.g., see comments for l.260-261 and l.280-281). Detailed comments below are indexed by line number.
l.22-25 “We reintroduced [provide number] mostly captive-bred conures in a protected area. The birds were acclimatized for at least five months at the release site, where they were trained to recognize native foods and develop foraging skills. Subsequently, we conducted a soft release, followed by daily supplementation.”
l.26 “For three years following the release, we recorded data on the conures’ diet…”
l.28 “The three most consumed species…” [note: chocolate is my favorite food, but it represents only a small part of my diet ;) ]
l.28-29 “Conures spent more time feeding on altered…”
l.30 “…natural feeding occurred during… their preferred plants…” [note: I am not sure what you mean by “natural feeding”]
l.31 “The conures’ incorporation of new species in their diet and the transition… occurred gradually”
l.33-34 “…immediately had more difficulty finding food.”
l.34 “can transition to a wild diet” [note: avoid using “adapt” in this context, since that word has a very specific meaning in biology, referring to the evolutionary process, which is not what you are talking about here]
l.40 “affect species, and for designing effective…”
l.41-43 “Reintroductions have been implemented to restore populations to areas from which they had been extirpated, and these initiatives have become more common as a result of their media appeal and symbolic meaning as a correction for human-driven defaunation.”
l.45 “However, many reintroductions are unsuccessful dues to…”
l.55 Remove the paragraph break here. “Brazil has both the largest number…”
l.57-58 “parrots are likely to face even more severe declines…”
l.58 “Despite this dire outlook, parrots are…”
l.64 “and even a species that is extinct in the wild (Cyanopsitta spixii).”
l.66-67 “information on their habitat and foraging requirements, which are necessary for supporting reintroduction efforts and …”
l.68 Instead of “exhuberant” I would suggest “spectacular” or “striking”
l.69 Instead of “lush” I would suggest “spectacular” or “striking” (whichever one you didn’t use for l.68)
l.72 “it is listed”
l.73 “despite its symbolic”
l.74 lowercase “c” for “conure’s”
l.77 “estimated its population size”
l.77 “on its diet” [note: “the species” is singular, “the conures’ ” is plural]
l.78 “which are mostly known from opportunistic”
l.79 “from captivity was implemented”
l.83 “conserving key food resources and habitats”
l.85-86 “post-release transition from…” [see comment for l.34]
l.91 either “of Pará state” or “of the state of Pará”
l.91 “The protected area of 13.93 km2 is connected to another..” [note: decimal point instead of a comma]
l.92 “and south, but is also bordered by urban areas to the north and west.”
l.94 “as well as altered landscapes and…”
l.95 “resulting from…”
l.96 “large bodies of water”
l.97-98 “forest, the area is seasonal, with a rainy season (December to May) and a dry season (June to November).”
l.100 “743 mm per month.”
l.101 “at the Lymington Foundation”
l.102 “a conservation organization that is dedicated to preserving Brazil’s native flora and fauna, and which operates a captive breeding facility for parrots.”
l.102 “At the LF, groups…”
l.104 “They were also screened for common…”
l.105-106 “local regulations governing animal translocations.”
l.107 On what date did this occur?
l.108 “at the release site” “previously cleared site in secondary”
l.109 “The aviary structure is composed of two modules: a…”
l.110 “The conures were kept…”
l.111 “During the acclimatization…”
l.112 I am not quite sure what you mean by “presented to the local fauna”
l.116 “known to be included in their natural diet”
l.120 “develop a naturalistic foraging behavior.” “considered ready for release”
l.122 On what date did this occur? This is useful to provide a reference point for the subsequent observation period about which you report in this manuscript.
l.123 “release. We provided the birds with the option to return [to the aviary?]…”
l.123 What do you mean by “stimulated them to avoid dispersal”?
l.126 “commercially available fruits”
l.127 What do you mean by “ration”? do you mean “a commercial bird food”?
l.127 “in order to provide the released birds with a continuous supply of food.”
l.129 “to the aviary, serving as social attractants.”
l.132 “conures was located and followed…”
l.133 “returned at the end of the day.”
l.133 What do you mean by “dormitories” here? Did the conures sleep in roost trees? Or in nest boxes? This would be interesting to describe a bit; since most parrots roost in trees overnight, except when they are nesting (and even then, both parents don’t always sleep in the nest cavity).
l.133-134 “When necessary, we used radio telemetry and playback calls to locate the birds. Once they were located, we…”
l.136 “with a 70-300mm zoom lens to record any feeding events.”
l.136-137 “For each feeding observation, we noted the plant species…”
l.139-140 “the location’s habitat type was classified according to habitat characteristics or location on the park map.”
l.146-147 “We were able to collect foraging data for 86 of the 164 observation campaigns that we conducted; in the remaining cases we lost track of the birds or could not reliably observe there foraging behavior.”
l.148-149 “observations; mean feeding session duration was 19 minutes.” Note: it would be good to provide the range and/or standard deviation for this mean.
l.151-152 “Some species were consumed much more frequently than others; the three most-used species (…) represented more than 77% of the…”
l.154 “most plant families”
l.157-158 “at Utinga State Park and its vicinity.”
l.158 Lowercase “r” for “refers”
l.165 “and fruits but were occasionally seen eating”
l.166-167 “They typically moved toward the terminal end of twigs and branches to pluck fruit, and the processed the fruit while holding it in one foot.”
l.170 Earlier you mentioned that the birds were offered native fruits in the aviary in order to train them to recognize it…. But you do not mention any “training sessions.” Since you mention them here, you should describe the training sessions above. OR, if you didn’t have specific training sessions, then re-word this sentence to something like “even before the release, when the birds were offered native foods in the aviary setting.”
l.172 “While the conures were sometimes observed eating the pulp of fruits, their most common feeding method was to peel…”
l.188 “would usually land…” (you can’t say “always” since the next sentence says that they sometimes went lower to access palm fruits under the canopy)
l.191 On the other hand, when feeding at the forest edge or in patches of secondary vegetation, they…”
l.194 You could delete the paragraph break here.
l.203 “in the vicinity of”
l.204 “fall to the ground”
l.212 “until resuming their search for food.”
l.213 “around 18:00, though on days…”
l.218 “The increase in species diversity in the diet of the reintroduced…” [note: the word “richness” can also imply that you are talking about nutrient/energy content, so it’s best to be specific here]
l.219-220 “The four most consumed…” “to be recorded following the birds’ release.”
l.223 “but were rapidly found by the conures…”
l.224 “in the wild. Four species frequently…”
l.228-229 “Conures following the date of release, as documented during post-release monitoring campaigns.”
l.232 “During the first months after release…”
l.233 “feeders, and during the first year conures typically spent over 70% of their foraging time feeding on supplemental food.”
l.238 “their feeding behavior quickly converged on what…” [see note for l.34]
l.240 “separating from the group in order to return and feed on supplemental”
l.246 “after release was observed to return alone…” [note: how do you define “dispersed” in your study?]
l.248 “even immediately after release from the aviary.” [see previous note]
l.250 “We recaptured three individuals to prevent their starvation, attracting them with (…) seeds. The fourth individual entered…”
l.252-254 “Another individual dispersed after spending one year with the group, and was resighted [how many?] months later with a group of…”
l.254-256 Delete this sentence; since you do not have any direct observations, you should avoid speculating or assuming anything.
l.257 “fed mostly in vegetation…”
l.258-259 “Golden conures spent most (almost 80%) of their foraging in terra firma forest (Table 2), and only about 20% of their time foraging in primary forest (Firgure 6a).”
l.260-261 You need to explain more clearly what these percentages mean… Do you mean that most of your observations were made during the rainy season? Otherwise, you need to standardize the feeding time by the total observation time. i.e., number of minutes feeding/number of minutes observed. Ideally you would also indicate what the birds were doing when they were not feeding (perched, flying, etc.)
l.262 “throughout the year.”
l.262-263 “Although E.oleracea’s fruit production is greater during the wet season, conures…”
l.263-266 “The conures consistently spent much less time feeding in primary vegetation than in secondary vegetation, but during the dry season the percentage of time feeding in primary vegetation nearly doubled (Figures 7a,b).”
l.270 Why do you say “explored” instead of “used” here?
l.280-281 Fig.6 See comment for l.260-261. If you do not standardize the data, then the differences in time feeding could simply reflect the amount of time spent observing the birds (i.e., even if the birds spent 50% of their time feeding during all seasons, if you observed them for more time in the rainy season, you would record more feeding time for the rainy season.)
l.280-281 I would delete “physiognomies” here; “primary or secondary vegetation” is sufficient.
l.286 see previous comment
l.290-291 “The number of species consumed by reintroduced…within the range reported by Renton…”
l.293 “diet have also been reported…”
l.295 “account for only…” [note: use a decimal point, not a comma]
l.296 “accounted for larger share of the records (83.4%) [note: use a decimal point, not a comma]
l.296 delete “demonstrating (…) diet.” This statement is circular and therefore uninformative: “favorite” species are by definition those that are consumed the most, and so their relative importance is (by definition) the greatest.
l.298 “which was the most frequently consumed food item in our study,” (see comment for l.28)
l.299 Need to clarify wording here: Did Laranjeiras made observations of species feeding on B.crassifolia? or did he make observations of Golden Conures foraging?
l.301 “and we recorded the conures feeding on it when the first unripe fruits appear in September.”
l.301-302 “These observations indicate this fruit’s…” (note: strictly speaking your evidence is observational and not experimental, so “demonstrate” is too strong a word here)
l.303 “is likely important for supporting a Golden Conure population.” (see previous comment)
l.307-308 “which was consistently consumed throughout the year in our study, was previously reported by Sick [18] but no other published reports.”
l.309-310 “which is found throughout the Amazon, is present at high densities in urban environments and forests.” (note: I am not sure how the commercial value is relevant here. Do you mean to state that it is planted in urban environments due to its commercial value? But in that case, how does its commercial value explain its high density in forests? This needs to be clarified, or else delete “for its commercial value”)
l.310-312 “Acai fruit was one of the food items that was most often provided during food training, and this might explain why it was so frequently consumed post-reintroduction.”
l.313 “with six species consumed by the conures.”
l.313-315 “Palms, which produce fat-rich fruits that are available year-round, are thought to be essential food sources for parrots, many of whom consume fruits from multiple palm species.”
l.318 “dispersal of palm fruits”
l.319 “but only rarely with other species.”
l.319 “These dispersal events mainly occurred when…”
l.322 “with other psittacines. The latter were also seen…”
l.324-325 “As reported for other studies of parrots, we observed that conures wasted a great deal of food during processing and feeding. While this could be explained by a high degree of selectivity for particular plant parts, it could also result from inefficient food processing and frequent dropping of food while feeding.”
l.327 I am not sure why similar feeding behavior patterns (i.e. dropping lots of food) in captivity would support one hypothesis over the other. Why don’t you expect captive birds to be selective?
l.328-329 I don’t follow your logic here: even if food is abundant, the birds might be selective. An selectivity doesn’t necessarily reflect caution, it could reflect different energy/nutrient content of different plant parts.
l.333 see comment for l.34 Most likely this is an instinctive behavior. Did you observe any vigilance in the aviary, pre-release?
l.337-338 “The order in which food-plant species were incorporated into the diet post-reintroduction was apparently influenced by pre-release training; species offered in the aviary were among the first to be consumed post-reintroduction.”
l.343 “Although the conures in our study were observed feeding on almost all…”
l.346 Insert a sentence here: “We should note that we were unable to collect Croton matourensis for food training and supplementation.”
l.347 “was offered daily”
l.348-349 Delete “while (…) supplementation.”
l.350-352 At some point you need to discuss the fact that your observations do not allow you to determine the complete list of what the conures are exploring or feeding on. Without observing them 100% of the time, your observations are necessarily incomplete.
l.352-354 Is the discovery rate slower? Or does the diet of Golden Conures appear to include more species, and so the curve takes longer to flatten even if the discovery rate is the same?
l.356-357 “constant supplementation as an incentive to keep…”
l.362 “days after reintroduction” … “the free-living conures”
l.363-365 “can be difficult for some individuals who are unable to forage successfully, especially when they become separated from the group. Although we do not know for certain whether…”
l.368 “safer and more successful transition the aviary to the wild.”
l.369 Remove the paragraph break here.
l.369-370 “a safe and reliable foraging option for reintroduced Golden Conures.” Note: I would delete the rest of the sentence (“which was… and starvation”)
l.371-372 “The importance of food supplementation has been highlighted by other studies, citing it as a critical factor contributing to…”
l.373-380 The logic of the first sentence is difficult to follow, and the rest of the paragraph is a bit cumbersome. A possible rewording: “Even though the release area was mostly covered by primary forest, the conures clearly preferred to forage in disturbed areas with secondary vegetation, presumably because the three most consumed species are found in the human-altered areas. Those species fruit primarily during the rainy season, which explains the greater use of disturbed areas during the wet season. When those fruits are not available during the dry season, the conures then shift to feeding more in primary vegetation habitats.”
l.383-384 “when the reintroduced birds have a higher chance of locating abundant food.”
l.389-390 “can adjust/acclimatize to a natural environment given a supportive reintroduction process, and can develop…”
Comments on the Quality of English Language
The manuscript was generally easy to understand, but there were many minor grammatical mistakes and awkwardly worded phrases. I have provided detailed comments that should improve the English and clarify the wording throughout the text.
Author Response
The point by point response is in the attached document.

Reviewer 3 Report
Comments and Suggestions for Authors
This is a very well done, straight forward paper describing the diets of released Golden Parakeets in Brazil. The paper is very clear and well written and makes a valuable contribution to the field of parrot release science.
Overall comments:
I have one major concern that needs to be addressed before publication. The feeding observations are presented as total time seen feeding. Unfortunately, I am unable to find any quantification of the amount of effort across the study period or habitats. As a result, it is hard to interpret the Figures 3 and 6. While I realize that it may be impossible to quantify exactly how many hours were invested in radio tracking and following the birds, there needs to be some scaling of foraging time by effort. As a result, I suggest that the results be presented as # minutes per day of monitoring or something like that. This would also be useful as it would give the reader some idea of how much time per day the birds were foraging (or at least how much time per day they were seen foraging).
In addition, more detail on the monitoring is needed (around Line 130). Give the number of days per month that birds were tracked. Given the reality of field work, there is probably some variation in the monitoring effort over time and this should be reflected in the date presentation. State specifically that birds were found in the roost locations using telemetry and then followed from there. Also comment as to whether or not birds were usually successfully tracked throughout the day or at least qualitatively talk about how often they were lost, and researchers gave up etc. Just to give the readers better idea about how to interpret the data presented.
Taxonomy. Please use the common ornithological name in English: Golden Parakeet. The term “Conure” is normally used only in Aviculture for captive birds.
I think that this paper could put more emphasis on the role that providing foods pre release played in the ultimate diet. Two spots for this are L337 (see below) and by adding information to Table 2 on which species were given in captivity pre release (see comment in pdf).
Specific comments:
Table 2: It would be interesting to know all the types of areas the plants occurred in. For example are some species common in all habitats but only used in Secondary Terra Firme? Was there secondary flooded forest in the study area or just primary? This clearer explanation of the available habitats could be included around line 93.
Fig 6a: Remind the readers here that most of the habitat was primary forest and secondary was just a small percentage.
Line 302. No evidence is presented to suggest that Byrsonima is vital for species survival. There is a big difference between a favored food and a food vital for survival. Please remove this unless additional strong evidence is presented. A statement like this could be adopted by governments or others and be used to inappropriately interfere with species management.
L 337. You could show the time to first use of plant species they were trained on in Figure 4. For each time a new species they were trained on was consumed you could put an asterisk or open dot instead of the dark blue solid dot. This would make it clear that many of the species that were given pre release were consumed shortly after release. This would allow readers to better interpret the point you are making on 337 and make it a stronger point.
Comments on the Quality of English Language
There are a few places where the adjectives are a bit too exaggerated, but overall it is very well written.
Author Response
The point by point response is in the attached document
